# Dearomatization of 3-Aminophenols for Synthesis of Spiro[chromane-3,1′-cyclohexane]-2′,4′-dien-6′-ones via Hydride Transfer Strategy-Enabled [5+1] Annulations

**DOI:** 10.3390/molecules29051012

**Published:** 2024-02-26

**Authors:** Jia-Cheng Ge, Yufeng Wang, Feng-Wei Guo, Xiangyun Kong, Fangzhi Hu, Shuai-Shuai Li

**Affiliations:** 1College of Chemistry and Pharmaceutical Sciences, Qingdao Agricultural University, Qingdao 266109, China; gejiacheng@hailir.cn (J.-C.G.); wyf1758561298@163.com (Y.W.); guofengwei2017@163.com (F.-W.G.); kongxiangyun528@126.com (X.K.); 2Hailir Pesticides and Chemicals Group Co., Ltd., Qingdao 266109, China

**Keywords:** dearomatization, spiro[chromane-3,1′-cyclohexane]-2′,4′-dien-6′-ones, C(sp^3^)–H bond functionalization, antifungal activity

## Abstract

The Sc(OTf)_3_-catalyzed dearomative [5+1] annulations between readily available 3-aminophenols and *O*-alkyl *ortho*-oxybenzaldehydes were developed for synthesis of spiro[chromane-3,1′-cyclohexane]-2′,4′-dien-6′-ones. The “two-birds-with-one-stone” strategy was disclosed by the dearomatization of phenols and direct α-C(sp^3^)–H bond functionalization of oxygen through cascade condensation/[1,5]-hydride transfer/dearomative-cyclization process. In addition, the antifungal activity assay and derivatizations of products were conducted to further enrich the utility of the structure.

## 1. Introduction

Aromatic compounds as bulk and fundamental chemical feedstocks play a prominent role in organic synthesis [1,2,3]. Dearomatization is the high-value-added transformation of aromatic compounds to generate structurally diverse three-dimensional polycyclic molecules [4,5,6,7]. Due to the high significance of dearomatization in assembling sophisticated polycyclic architectures with enhanced sp^3^-character, much more attention has been paid to dearomatization chemistry [4,5,6,7,8,9,10,11,12,13,14,15,16]. Among the dearomatization reactions, phenol is one of the most studied structures which could be facilely converted into cyclohexadienone, and many strategies have been developed to achieve dearomatization of phenols [13,14,15,16]. For example, *ortho*-substituted phenols were commonly used to undergo oxidant-promoted oxidative dearomatization or transition-metal-catalyzed allylation, alkylation, amination to generate cyclohexadienones (Figure 1a) [17,18,19]. Although great progress has been made in these transformations, the *ortho*-substituted phenols have to be prefabricated, which brings limitations in practical production. Notably, the employment of phenols and dielectrophiles as starting materials for direct dearomatization of phenols was a highly atom- and step-economic strategy (Figure 1b). Until now, the dearomatization of phenols by employment of the phenols without pre-installed substituents at the dearomatization site via direct formation of two chemical bonds was a huge challenge.

Hydride transfer/cyclization has attracted intense interest as an efficient strategy with the breaking and formation of a number of bonds in one operation for construction of privileged molecules [20,21,22,23,24,25,26,27,28,29,30,31,32,33,34,35,36,37,38]. Among the diverse transformations, the α-C(sp^3^)−H bond functionalization of amine occupied the majority (Figure 1c) [23,24,25,26,27,28,29,30,31,32,33,34,35], while the application of an α-C(sp^3^)−H bond of oxygen as hydride donor was relatively challenging, which might be attributed to (1) lower reactivity of the α-C(sp^3^)−H bond of oxygen than that of *tert*-amine; (2) the in situ-formed acyclic oxocarbenium intermediate tends to hydrolyze [36,37,38]. This disadvantage led to few acceptors being available for initiating the α-C(sp^3^)−H bond functionalization of oxygen (Figure 1d).

Aromatization is an important thermodynamic driving force for organic transformation [39,40,41,42]. Inspired by the above-mentioned challenge, we designed to utilize the in situ-assembled *ortho*-quinone methides (*o*-QMs) from phenols and aldehydes as hydride acceptors for driving hydride transfer with the potent propensity of aromatization. In addition, the formal [5+1] annulation that undergo dearomatization/rearomatization/dearomatization process would achieve the direct dearomatization of phenols and α-C(sp^3^)−H bond functionalization of oxygen (Figure 1e).

As a continuation of our interest in developing hydride transfer-involved reactions for the rapid construction of privileged heterocyclic skeletons [7,22,34,35,42], herein we report the “two-birds-with-one-stone” strategy to dearomatize phenols and functionalize α-C(sp^3^)−H bond of oxygen. This dearomative [5+1] annulation provided a variety of cy spiro[chromane-3,1′-cyclohexane]-2′,4′-dien-6′-ones from 3-aminophenols and *O*-alkyl *ortho*-oxybenzaldehydes through Sc(OTf)_3_-catalyzed cascade condensation/[1,5]-hydride transfer/dearomative-cyclization process (Figure 1f). It is worth mentioning that the direct dearomatization of phenols combined with the functionalization of an α-C(sp^3^)−H bond of oxygen via a hydride transfer strategy was achieved unprecedentedly. In addition, the antifungal activity assay preliminarily showed the potential of these products in the prevention and control of agricultural pathogens.

## 2. Results and Discussion

At the outset of the reaction, the *O*-benzyl salicylaldehyde-derived substrate **1a** and phenol **2a** were selected as model substrates to examine the reaction (Table 1). Firstly, Lewis acids Sc(OTf)_3_, Mg(OTf)_2_, and Zn(OTf)_2_ were applied as catalysts in DCE to promote the desired transformation. To our delight, the expected cyclohexadienone-fused spirochromane **3a** was obtained through the Lewis acids-catalyzed cascade condensation/[1,5]-hydride transfer/dearomative-cyclization, and a 52% yield was isolated with Sc(OTf)_3_ as catalyst (entry 1). Next, various Brønsted acids were also applied to evaluate the reaction. As shown, Brønsted acids showed slightly weaker catalytic activity than Lewis acids (entries 4–7). Subsequently, diverse solvents were investigated with Sc(OTf)_3_ as catalyst to further improve the yield of the product. Delightedly, fluorinated alcohol TFE (2,2,2-trifluoroethanol) was efficient in mediating the dearomatization/aromatization-dearomatization process to afford the product **3a** in 84% yield and excellent diastereoselectivity (entries 8–13). Moreover, HFIP (1,1,1,3,3,3-Hexafluoro-2-propanol) was also a good reaction medium for the transformation (entry 14). Then, the ratios of substrates, reaction temperature, and the loading of catalyst were meticulously screened. The results showed that the adjustment of the ratio of **1a** and **2a** and reaction temperature was unavailing for improving the reaction efficiency (entries 15–19). The examination of the loading of catalyst indicated that 20 mol% of Sc(OTf)_3_ was the suitable dosage for the reaction (entries 20–22). At last, the optimal reaction conditions were determined to be those described in entry 21 of Table 1.

With the optimal reaction conditions in hand, the generality of the dearomative [5+1] annulation with respect to diverse *O*-alkyl *ortho*-oxybenzaldehydes **1** was investigated (Figure 2). Notably, substrates **1** carrying electron-withdrawing or -donating groups on the phenyl ring of the benzyl moiety were applicable for the [5+1] annulation with phenol **2a** to afford the corresponding products **3a**–**g** in moderate to excellent yields and excellent diastereoselectivities. Moreover, among the diverse substituents, the electron-withdrawing group had some effect on the reaction efficiency, delivering slightly lower yields of products (**3b**, **3c**, **3f**, **3g**). Apart from 2-phenyl spirochromane, the 2-naphthyl spirochromane **3h** could also be obtained in 90%. Moreover, isopropyl could act as hydride donor to conduct the hydride transfer-involved dearomative [5+1] annulation to provide the gem-dimethyl-substituted spirochromane **3i**. In addition, the methyl-substituted benzyl and allyl-derived *ortho*-oxybenzaldehydes **1** were good reaction partners with phenol to perform the dearomative [5+1] annulation for giving the diversity-enriched spiro[chromane-3,1′-cyclohexane]-2′,4′-dien-6′-ones **3j** and **3k** in 70% and 90% yields, respectively.

Next, further investigation of the substrate scope with regard to phenols was performed for dearomatization of diverse phenols (Figure 3). Various types of phenols were applied to react with *O*-alkyl *ortho*-oxybenzaldehydes **1**. For example, non-substituted phenol was unavailable for the transformation, and electron-rich sesamol gave several unascertained products. Moreover, *m*-aminophenols showed excellent activity for reacting with *O*-alkyl *ortho*-oxybenzaldehydes. For instance, dimethylamine, pyrrolidine, piperidine, and azepine-substituted phenols were applicable for the transformation, affording the corresponding spiro[chromane-3,1′-cyclohexane]-2′,4′-dien-6′-ones **3l**-**o** in moderate to excellent yields. In addition, dibenzylamine- and diallylamine-substituted phenols were also available for the reaction which further enriched the diversity of the substituents of products. The wide tolerance of the substituents on the phenols made the product more feasible for late-stage functionalization. On the other hand, the *ortho tert*-butyl could be replaced by *iso*-propyl or *sec*-butyl, delivering the corresponding products **3t** or **3u** in slightly lower yields than product **3a**. 

Then, the practicality of this protocol was demonstrated by the gram-scale synthesis of products **3**. As shown in Figure 4a, the reaction efficiency was almost unaffected in a 4 mmol scale to give the corresponding products **3a** and **3k** in 67% and 74% yields, respectively (Figure 4a). In addition, in order to illustrate the biological activity of the product, the antifungal activity assay was performed (See Appendix A). Compounds **3a**, **3j**, **3k**, **3q**, **3t** and **3u** were evaluated for their antifungal activities against four economically important phytopathogenic fungi: *Rhizoctonia solani*, *Alternaria solani*, *Alternaria mali*, and *Sclerotium rolfsii*. The results showed that most of the tested compounds possessed in vitro antifungal activity at a concentration of 200 mg/L. Especially, compound **3a** exhibited remarkable antifungal potency among all target compounds, with inhibition rates of 40.21, 60.35, 53.56 and 29.09% at a concentration of 200 mg/L, respectively, against *Rhizoctonia solani*, *Alternaria solani*, *Alternaria mali*, and *Sclerotium rolfsii*. The results preliminarily showed the potential of these products in the prevention and control of agricultural pathogens. Moreover, the selective reduction in the terminal alkene by H_2_/Pd/C was performed to provide the 2-ethyl spirochromane **4k** in 82% yield (Figure 4b). This transformation remedied the deficiency that was unable to furnish 2-ethyl spirochromane by the direct dearomative [5+1] annulation. Apart from spirochromanes, polyarylated methane **5a** could also be provided by twice nucleophilic addition with the employment of the *O*-alkyl *ortho*-oxybenzaldehyde **1v** (Figure 4c). This transformation indicated that the latter nucleophilic addition and hydride transfer were competitive reaction pathways.

Subsequently, the key factor for the hydride transfer/dearomative-cyclization was investigated (Figure 5). In a hydride transfer reaction, the distance between hydride donor and hydride acceptor was decisive for the occurrence, and the conclusion was verified by the investigation on the “buttressing effect”. As shown in Figure 5a, remarkable enhancement of the reactivity by the bulky group *ortho* to the alkoxy group could be clearly observed, which might be due to the steric repulsion between the ortho group and the alkoxy group shortening the distance between hydride donor and hydride acceptor. Then, the investigation into the effect of hydride donors demonstrated that the transfer ability of the hydride donors was a key factor as well (Figure 5b). The examination of the α-C(sp^3^)−H bond of ethyl, *iso*-propyl, and benzyl adjacent to oxygen showed the difference in the activity, which might be dependent on the transfer ability of the hydride donors and the stability of the cations generated upon hydride transfer.

Based on the above experiments and precedent reports [37,38], a plausible mechanism was proposed to rationalize the dearomative [5+1] annulation (Figure 6). First, the catalyst scandium-aggregated *O*-alkyl *ortho*-oxybenzaldehyde **1** and phenol **2** to mediate the Friedel–Crafts hydroxyalkylation condensation. Then, the adduct dehydrated immediately to generate an *o*-QM intermediate **II**. The propensity for aromatization of *o*-QM as a driving force initiated the α-hydride of oxygen transfer to yield the zwitterionic intermediate **III**. Next, the dearomative cyclization of **III** took place to furnish the cyclohexadienone-fused spirochromane **3**.

## 3. Materials and Methods

### 3.1. General Information

Unless otherwise noted, all reagents and solvents were purchased from the commercial sources (from Adamas-beta, Shanghai, China) and used as received. Thin layer chromatography (TLC) was used to monitor the reaction on a Merck 60 F254 precoated silica gel plate (0.2 mm thickness). TLC spots were visualized by UV-light irradiation on a Spectroline Model ENF-24061/F 254 nm. The products were purified by flash column chromatography (200–300 mesh silica gel) eluted with the gradient of petroleum ether and ethyl acetate. Proton nuclear magnetic resonance spectra (^1^H NMR) were recorded on a Bruker 500 MHz or 400 MHz NMR spectrometer (CDCl_3_, DMSO-*d_6_* or Methanol-*d*_4_ solvent). The chemical shifts were reported in parts per million (ppm), downfield from SiMe_4_ (δ 0.0) and relative to the signal of chloroform-*d* (δ 7.26, singlet), dimethyl sulfoxide-*d*_6_ (δ 2.54, singlet) or methanol-*d*_4_ (δ 3.31, quintuplet). Multiplicities were afforded as: s (singlet); d (doublet); t (triplet); q (quartet); dd (doublets of doublet) or m (multiplets). The number of protons for a given resonance is indicated by nH. Coupling constants were reported as a *J* value in Hz. Carbon nuclear magnetic resonance spectra (^13^C NMR) were referenced to the appropriate residual solvent peak. High-resolution mass spectral analysis (HRMS) was performed on Waters XEVO G2 Q-TOF. The *ortho*-substituted benzaldehydes were prepared according to the literature [43].

### 3.2. General Procedure for the Dearomative [5+1] Annulation

A sealed tube was charged with *O*-alkyl *ortho*-oxybenzaldehyde **1** (0.1 mmol), phenol **2** (0.15 mmol), Sc(OTf)_3_ (20 mol%), and TFE (1.0 mL). The mixture was stirred at 120 °C for 5 h. Upon completion of the reaction, as indicated by TLC analysis, the mixture was concentrated in vacuum and the residue was directly purified by flash column chromatography on silica gel (eluent: ethyl acetate/petroleum ether, 1:3) to afford the desired spiro[chromane-3,1′-cyclohexane]-2′,4′-dien-6′-ones **3a**–**u**.

### 3.3. Gram-Scale Synthesis and Derivatization of Products ***3***

A round-bottom flask was charged with *O*-alkyl *ortho*-oxybenzaldehydes **1a** (4.0 mmol, 1.07 g) or **1k** (4.0 mmol, 0.87 g), phenol **2a** (6.0 mmol, 0.99 g), Sc(OTf)_3_ (20 mol%), and TFE (40.0 mL). The mixture was stirred at 120 °C for 5 h. Upon completion of the reaction, as indicated by TLC analysis, the mixture was concentrated in vacuum and the residue was directly purified by flash column chromatography on silica gel (eluent: ethyl acetate/petroleum ether, 1:3) to afford the desired spiro[chromane-3,1′-cyclohexane]-2′,4′-dien-6′-ones **3a** in 67% yield or **3k** in 74% yield, respectively.

A reaction tube was charged with cyclohexadienone-fused spirochromane **3k** (0.1 mmol, 36.6 mg) and 30% by wt Pd/C (10% by wt relative to **3k**) in 1.0 mL of MeOH. The tube was equipped with a magnetic stir bar, and the suspension was sealed with a septum under an atmosphere of H_2_ supplied via a balloon for 6 h. Upon completion of the reaction, as indicated by TLC analysis, the suspension was filtered through a pad of Celite. The filtrate was concentrated in vacuum. The residue was directly purified by flash column chromatography on silica gel (ethyl acetate: petroleum ether, 1:3) to give the desired product **4k** in 82% yield and 1:1 diastereoselectivity.

### 3.4. The Procedure for Synthesis of Product ***5a***

A tube was charged with *O*-alkyl *ortho*-oxybenzaldehyde **1v** (0.1 mmol, 32.8 mg), phenol **2a** (0.22 mmol, 36.4 mg), Sc(OTf)_3_ (20 mol%), and TFE (1.0 mL). The mixture was stirred at room temperature for 5 h. Upon completion of the reaction, as indicated by TLC analysis, the mixture was concentrated in vacuum and the residue was directly purified by flash column chromatography on silica gel (eluent: ethyl acetate/petroleum ether, 1:15) to afford the polyarylated methane **5a** in 81% yield.

### 3.5. Investigation on the “Buttressing Effect”

A tube was charged with *O*-alkyl *ortho*-oxybenzaldehydes **1** bearing methyl, isopropyl, or tertbutyl (0.1 mmol), phenol **2** (0.15 mmol), Sc(OTf)_3_ (20 mol%), and TFE (1.0 mL). The mixture was stirred at 120 °C for 5 h. Upon completion of the reaction, as indicated by TLC analysis, the mixture was concentrated in vacuum and the residue was directly purified by flash column chromatography on silica gel (eluent: ethyl acetate/petroleum ether, 1:3) to afford the desired spiro[chromane-3,1′-cyclohexane]-2′,4′-dien-6′-ones **3v**, **3t**, or **3a**. A remarkable enhancement of the reactivity by the bulky group ortho to the alkoxy group could be clearly observed, which might be due to the steric repulsion between ortho group and alkoxy group shortening the distance between hydride donor and hydride acceptor.

### 3.6. Investigation on the Effect of Hydride Donors

A tube was charged with *O*-ethyl, *O*-isopropyl, or *O*-benzyl *ortho*-oxybenzaldehydes **1** (0.1 mmol), phenol **2** (0.15 mmol), Sc(OTf)_3_ (20 mol%), and TFE (1.0 mL). The mixture was stirred at 120 °C for 5 h. Upon completion of the reaction, as indicated by TLC analysis, the mixture was concentrated in vacuum and the residue was directly purified by flash column chromatography on silica gel (eluent: ethyl acetate/petroleum ether, 1:3) to afford the desired spiro[chromane-3,1′-cyclohexane]-2′,4′-dien-6′-ones **3w**, **3i**, or **3a**. A remarkable enhancement of the reactivity by the bulky group ortho to the alkoxy group could be clearly observed, which might be due to the steric repulsion between ortho group and alkoxy group shortening the distance between hydride donor and hydride acceptor. The examination of the α-C(sp^3^)−H bond of ethyl, *iso*-propyl, and benzyl adjacent to oxygen showed the difference in the activity, which might be dependent on the transfer ability of the hydride donors and the stability of the cations generated upon hydride transfer.

### 3.7. Characterization of Products

8-(tert-butyl)-4′-(diethylamino)-2-phenylspiro[chromane-3,1′-cyclohexane]-2′,4′-dien-6′-one (**3a**). Flash column chromatography on a silica gel (ethyl acetate: petroleum ether, 1:3) afforded the product (35.3 mg, 85% yield) as a white solid.

^1^H NMR (500 MHz, CDCl_3_) δ 7.30 (m, 2H), 7.17–7.08 (m, 4H), 6.95 (d, *J* = 7.4 Hz, 1H), 6.82 (d, *J* = 7.6 Hz, 1H), 6.56 (d, *J* = 10.4 Hz, 1H), 6.26 (dd, *J* = 10.4, 2.1 Hz, 1H), 5.24 (s, 1H), 4.93 (d, *J* = 2.0 Hz, 1H), 3.85 (d, *J* = 16.5 Hz, 1H), 3.16–3.01 (m, 4H), 2.62 (d, *J* = 16.6 Hz, 1H), 1.30 (s, 9H), 0.93 (t, *J* = 7.0 Hz, 6H). ^13^C NMR (125 MHz, CDCl_3_) δ 195.4, 156.2, 152.6, 142.2, 137.9, 137.1, 127.9, 127.6, 127.1, 124.4, 121.0, 120.4, 119.0, 97.4, 83.5, 49.9, 45.0, 36.9, 34.8, 29.9. HRMS (ESI): [M+Na]^+^ calcd. for C_28_H_33_NNaO_2_^+^: 438.2404, found: 438.2408.

8-(tert-butyl)-4′-(diethylamino)-2-(2-fluorophenyl)spiro[chromane-3,1′-cyclohexane]-2′,4′-dien-6′-one (**3b**). Flash column chromatography on a silica gel (ethyl acetate: petroleum ether, 1:3) afforded the product (20.4 mg, 47% yield) as a yellow solid.

^1^H NMR (500 MHz, CDCl_3_) δ 7.41 (t, *J* = 6.9 Hz, 1H), 7.21 (d, *J* = 6.7 Hz, 1H), 7.18 (d, *J* = 7.7 Hz, 1H), 7.02 (m, 2H), 6.99–6.94 (m, 1H), 6.89 (t, *J* = 7.6 Hz, 1H), 6.71 (d, *J* = 10.4 Hz, 1H), 6.43 (dd, *J* = 10.4, 1.8 Hz, 1H), 5.61 (s, 1H), 5.05 (d, *J* = 1.4 Hz, 1H), 3.97 (d, *J* = 16.6 Hz, 1H), 3.21 (d, *J* = 6.2 Hz, 4H), 2.70 (d, *J* = 16.6 Hz, 1H), 1.34 (s, 9H), 1.04 (t, *J* = 6.7 Hz, 6H). ^13^C NMR (125 MHz, CDCl_3_) δ 194.5, 160.2 (d, *J* = 248.7 Hz), 156.0, 152.5, 142.2, 137.8, 129.5 (d, *J* = 3.8 Hz), 129.3 (d, *J* = 8.8 Hz), 127.9, 124.5 (d, *J* = 12.5 Hz), 124.4, 122.6 (d, *J* = 3.8 Hz), 120.8, 120.5, 119.0, 115.3, 115.2, 96.7, 77.6, 49.2, 44.9, 36.8, 34.7, 29.7. HRMS (ESI): [M+Na]^+^calcd. for C_28_H_32_FNNaO_2_^+^: 456.2309, found: 456.2311.

8-(tert-butyl)-2-(3-chlorophenyl)-4′-(diethylamino)spiro[chromane-3,1′-cyclohexane]-2′,4′-dien-6′-one (**3c**). Flash column chromatography on a silica gel (ethyl acetate: petroleum ether, 1:3) afforded the product (25.6 mg, 57% yield) as a white solid.

^1^H NMR (500 MHz, CDCl_3_) δ 7.40 (s, 1H), 7.28 (s, 1H), 7.23–7.19 (m, 2H), 7.17 (t, *J* = 7.7 Hz, 1H), 7.04 (d, *J* = 7.4 Hz, 1H), 6.91 (t, *J* = 7.6 Hz, 1H), 6.62 (d, *J* = 10.4 Hz, 1H), 6.40 (dd, *J* = 10.4, 2.2 Hz, 1H), 5.28 (s, 1H), 5.05 (d, *J* = 2.1 Hz, 1H), 3.92 (d, *J* = 16.6 Hz, 1H), 3.22 (d, *J* = 6.4 Hz, 4H), 2.71 (d, *J* = 16.7 Hz, 1H), 1.39 (s, 9H), 1.05 (t, *J* = 7.0 Hz, 6H). ^13^C NMR (125 MHz, CDCl_3_) δ 194.8, 156.1, 152.3, 141.7, 139.1, 137.9, 132.8, 128.4, 127.9, 127.7, 127.3, 126.1, 124.5, 120.8, 120.6, 119.3, 97.3, 82.7, 49.7, 45.0, 36.5, 34.7, 29.8. HRMS (ESI): [M+Na]^+^ calcd. for C_28_H_32_ClNaNO_2_^+^: 472.2014, found: 472.2010.

8-(tert-butyl)-4′-(diethylamino)-2-(m-tolyl)spiro[chromane-3,1′-cyclohexane]-2′,4′-dien-6′-one (**3d**). Flash column chromatography on a silica gel (ethyl acetate: petroleum ether, 1:3) afforded the product (38.2 mg, 89% yield) as a white solid.

^1^H NMR (500 MHz, CDCl_3_) δ 7.16 (m, 3H), 7.08 (t, *J* = 7.5 Hz, 1H), 7.00 (d, *J* = 7.3 Hz, 2H), 6.86 (t, *J* = 7.6 Hz, 1H), 6.61 (d, *J* = 10.4 Hz, 1H), 6.32 (dd, *J* = 10.4, 2.2 Hz, 1H), 5.25 (s, 1H), 4.99 (d, *J* = 2.2 Hz, 1H), 3.90 (d, *J* = 16.6 Hz, 1H), 3.28–3.01 (m, 4H), 2.67 (d, *J* = 16.6 Hz, 1H), 2.27 (s, 3H), 1.36 (s, 9H), 0.99 (t, *J* = 7.1 Hz, 6H). ^13^C NMR (125 MHz, CDCl_3_) δ 195.2, 156.1, 152.5, 142.1, 137.7, 136.9, 136.2, 128.2, 128.2, 127.8, 126.8, 124.5, 124.3, 120.8, 120.2, 118.8, 97.3, 83.4, 49.7, 44.9, 36.8, 34.7, 29.7, 21.4. HRMS (ESI): [M+Na]^+^ calcd. for C_29_H_35_NNaO_2_^+^: 452.2560, found: 452.2558.

8-(tert-butyl)-4′-(diethylamino)-2-(p-tolyl)spiro[chromane-3,1′-cyclohexane]-2′,4′-dien-6′-one (**3e**). Flash column chromatography on a silica gel (ethyl acetate: petroleum ether, 1:3) afforded the product (39.1 mg, 91% yield) as a white solid.

^1^H NMR (500 MHz, CDCl_3_) δ 7.29 (d, *J* = 1.9 Hz, 1H), 7.27 (s, 1H), 7.22–7.16 (m, 1H), 7.04 (m, 3H), 6.89 (t, *J* = 7.6 Hz, 1H), 6.65 (d, *J* = 10.4 Hz, 1H), 6.36 (dd, *J* = 10.4, 2.2 Hz, 1H), 5.30 (s, 1H), 5.02 (d, *J* = 2.2 Hz, 1H), 3.93 (d, *J* = 16.6 Hz, 1H), 3.20 (d, *J* = 6.7 Hz, 4H), 2.70 (d, *J* = 16.6 Hz, 1H), 2.30 (s, 3H), 1.40 (s, 9H), 1.03 (t, *J* = 7.1 Hz, 6H). ^13^C NMR (125 MHz, CDCl_3_) δ 195.3, 156.1, 152.6, 142.2, 137.7, 137.0, 134.1, 127.8, 127.6, 127.3, 124.3, 120.8, 120.2, 118.9, 97.3, 83.3, 49.8, 44.85, 36.8, 34.7, 29.7, 21.1. HRMS (ESI): [M+Na]^+^ calcd. for C_29_H_35_NNaO_2_^+^: 452.2560, found: 452.2556.

8-(tert-butyl)-4′-(diethylamino)-2-(4-fluorophenyl)spiro[chromane-3,1′-cyclohexane]-2′,4′-dien-6′-one (**3f**). Flash column chromatography on a silica gel (ethyl acetate: petroleum ether, 1:3) afforded the product (34.6 mg, 80% yield) as a yellow solid.

^1^H NMR (500 MHz, CDCl_3_) δ 7.40–7.31 (m, 2H), 7.18 (dd, *J* = 7.9, 1.7 Hz, 1H), 7.02 (dd, *J* = 7.5, 1.6 Hz, 1H), 6.95–6.85 (m, 3H), 6.62 (d, *J* = 10.5 Hz, 1H), 6.36 (dd, *J* = 10.5, 2.3 Hz, 1H), 5.30 (s, 1H), 5.02 (s, 1H), 3.89 (dt, *J* = 16.5, 1.1 Hz, 1H), 3.20 (q, *J* = 7.2 Hz, 4H), 2.69 (d, *J* = 16.6 Hz, 1H), 1.37 (s, 9H), 1.03 (t, *J* = 7.1 Hz, 6H). ^13^C NMR (100 MHz, CDCl_3_) δ 195.2, 162.3 (d, *J* = 244.0 Hz, 1H), 156.2, 152.5, 142.1, 137.9, 129.2 (d, *J* = 8.0 Hz, 1H), 127.9, 124.5, 120.9, 120.6, 119.2, 113.9 (d, *J* = 21.0 Hz, 1H), 97.3, 82.8, 49.8, 45.0, 36.9, 34.8, 29.8. HRMS (ESI): [M+Na]^+^calcd. for C_28_H_32_FNNaO_2_^+^: 456.2309, found: 456.2311.

2-(4-bromophenyl)-8-(tert-butyl)-4′-(diethylamino)spiro[chromane-3,1′-cyclohexane]-2′,4′-dien-6′-one (**3g**). Flash column chromatography on a silica gel (ethyl acetate: petroleum ether, 1:3) afforded the product (25.6 mg, 52% yield) as a white solid.

^1^H NMR (500 MHz, CDCl_3_) δ 7.28 (d, *J* = 8.5 Hz, 2H), 7.20 (s, 1H), 7.18 (s, 1H), 7.11 (d, *J* = 7.4 Hz, 1H), 6.95 (d, *J* = 7.4 Hz, 1H), 6.82 (t, *J* = 7.6 Hz, 1H), 6.53 (d, *J* = 10.4 Hz, 1H), 6.28 (dd, *J* = 10.4, 2.2 Hz, 1H), 5.20 (s, 1H), 4.95 (d, *J* = 2.1 Hz, 1H), 3.82 (d, *J* = 16.6 Hz, 1H), 3.14 (d, *J* = 5.4 Hz, 4H), 2.62 (d, *J* = 16.7 Hz, 1H), 1.29 (s, 9H), 0.97 (t, *J* = 7.1 Hz, 6H). ^13^C NMR (125 MHz, CDCl_3_) δ 195.7, 156.9, 153.1, 142.6, 138.6, 137.0, 131.0, 130.0, 128.6, 125.3, 122.3, 121.6, 121.4, 120.0, 98.1, 83.5, 50.5, 45.8, 37.5, 35.5, 30.6. HRMS (ESI): [M+Na]^+^ calcd. for C_28_H_32_BrNNaO_2_^+^: 516.1509, found: 516.1496.

8-(tert-butyl)-4′-(diethylamino)-2-(naphthalen-2-yl)spiro[chromane-3,1′-cyclohexane]-2′,4′-dien-6′-one (**3h**). Flash column chromatography on a silica gel (ethyl acetate: petroleum ether, 1:3) afforded the product (41.8 mg, 90% yield) as a white solid.

^1^H NMR (500 MHz, CDCl_3_) δ 7.86 (s, 1H), 7.84–7.80 (m, 1H), 7.80–7.76 (m, 1H), 7.70 (d, *J* = 8.5 Hz, 1H), 7.55 (dd, *J* = 8.5, 1.7 Hz, 1H), 7.45–7.41 (m, 2H), 7.22 (d, *J* = 7.0 Hz, 1H), 7.06 (d, *J* = 7.0 Hz, 1H), 6.92 (t, *J* = 7.6 Hz, 1H), 6.74 (d, *J* = 10.4 Hz, 1H), 6.34 (dd, *J* = 10.5, 2.2 Hz, 1H), 5.50 (s, 1H), 4.98 (d, *J* = 2.2 Hz, 1H), 3.98 (d, *J* = 16.6 Hz, 1H), 3.05 (s, 4H), 2.75 (d, *J* = 16.6 Hz, 1H), 1.41 (s, 9H), 0.83 (s, 6H). ^13^C NMR (125 MHz, CDCl_3_) δ 195.2, 156.1, 152.6, 142.1, 137.9, 134.8, 133.0, 132.5, 128.3, 127.9, 127.3, 126.6, 126.4, 125.8, 125.6, 125.5, 124.4, 120.9, 120.4, 119.1, 97.2, 83.43, 49.94, 44.8, 37.0, 34.8, 29.8. HRMS (ESI): [M+Na]^+^ calcd. for C_32_H_35_NNaO_2_^+^: 488.2560, found: 488.2550.

8-(tert-butyl)-4′-(diethylamino)-2,2-dimethylspiro[chromane-3,1′-cyclohexane]-2′,4′-dien-6′-one (**3i**). Flash column chromatography on a silica gel (ethyl acetate: petroleum ether, 1:3) afforded the product (22.8 mg, 62% yield) as a yellow oil.

^1^H NMR (500 MHz, CDCl_3_) δ 7.15 (dd, *J* = 7.8, 1.7 Hz, 1H), 7.00–6.92 (m, 1H), 6.82 (t, *J* = 7.6 Hz, 1H), 6.48 (d, *J* = 10.5 Hz, 1H), 6.42 (dd, *J* = 10.5, 2.3 Hz, 1H), 5.31 (d, *J* = 2.3 Hz, 1H), 3.69–3.58 (m, 1H), 3.39 (q, *J* = 7.1 Hz, 4H), 2.46 (d, *J* = 16.9 Hz, 1H), 1.47 (s, 3H), 1.39 (s, 9H), 1.33 (s, 3H), 1.22 (t, *J* = 7.1 Hz, 6H). ^13^C NMR (125 MHz, CDCl_3_) δ 195.9, 156.4, 151.1, 146.1, 138.1, 127.7, 124.3, 120.9, 119.6, 117.8, 97.4, 79.1, 77.2, 50.2, 44.9, 34.8, 33.8, 30.4, 29.7, 24.6, 21.6. HRMS (ESI): [M+Na]^+^ calcd. for C_24_H_33_NNaO_2_^+^: 390.2404, found: 390.2409.

8-(tert-butyl)-4′-(diethylamino)-2-methyl-2-phenylspiro[chromane-3,1′-cyclohexane]-2′,4′-dien-6′-one (**3j**). Flash column chromatography on a silica gel (ethyl acetate: petroleum ether, 1:3) afforded the product (30.0 mg, 70% yield) as a yellow oil.

^1^H NMR (500 MHz, CDCl_3_) δ 7.54–7.46 (m, 2H), 7.23 (dd, *J* = 7.9, 1.7 Hz, 1H), 7.20–7.12 (m, 3H), 7.03 (dd, *J* = 7.5, 1.6 Hz, 1H), 6.91 (t, *J* = 7.6 Hz, 1H), 6.50 (d, *J* = 10.5 Hz, 1H), 6.08 (dd, *J* = 10.5, 2.3 Hz, 1H), 5.08 (d, *J* = 2.3 Hz, 1H), 3.83 (d, *J* = 17.1 Hz, 1H), 3.09 (d, *J* = 7.4 Hz, 4H), 2.58 (d, *J* = 17.2 Hz, 1H), 1.86 (s, 3H), 1.45 (s, 9H), 0.93 (t, *J* = 7.1 Hz, 6H). ^13^C NMR (125 MHz, CDCl_3_) δ 195.2, 156.1, 150.7, 144.6, 142.3, 138.9, 127.9, 126.7, 126.7, 126.4, 124.7, 121.3, 120.4, 118.1, 98.5, 83.3, 50.9, 44.9, 34.9, 33.4, 30.2, 29.7, 20.1. HRMS (ESI): [M+Na]^+^ calcd. for C_29_H_35_NNaO_2_^+^: 452.2560, found: 452.2563.

8-(tert-butyl)-4′-(diethylamino)-2-vinylspiro[chromane-3,1′-cyclohexane]-2′,4′-dien-6′-one (**3k**). Flash column chromatography on a silica gel (ethyl acetate: petroleum ether, 1:5) afforded the product (32.9 mg, 90% yield) as a white solid.

^1^H NMR (500 MHz, CDCl_3_) δ 7.16 (d, *J* = 7.5 Hz, 1H), 6.95 (d, *J* = 7.3 Hz, 1H), 6.84 (t, *J* = 7.6 Hz, 1H), 6.58–6.42 (m, 2H), 5.86–5.71 (m, 1H), 5.51 (d, *J* = 17.2 Hz, 1H), 5.33 (d, *J* = 2.0 Hz, 1H), 5.19 (d, *J* = 10.8 Hz, 1H), 4.81 (d, *J* = 5.2 Hz, 1H), 3.67 (d, *J* = 16.4 Hz, 1H), 3.39 (dd, *J* = 14.1, 7.0 Hz, 4H), 2.61 (d, *J* = 16.4 Hz, 1H), 1.42 (s, 9H), 1.23 (t, *J* = 7.1 Hz, 6H). ^13^C NMR (125 MHz, CDCl_3_) δ 195.9, 156.6, 152.0, 143.1, 137.6, 133.3, 127.6, 124.4, 120.8, 120.1, 118.3, 117.2, 96.6, 80.8, 48.1, 44.9, 37.6, 34.7, 29.6. HRMS (ESI): [M+Na]^+^ calcd. for C_24_H_31_NNaO_2_^+^: 388.2247, found: 388.2244.

8-(tert-butyl)-4′-(dimethylamino)-2-phenylspiro[chromane-3,1′-cyclohexane]-2′,4′-dien-6′-one (**3l**). Flash column chromatography on a silica gel (ethyl acetate: petroleum ether, 1:3) afforded the product (34.8 mg, 90% yield) as a white solid.

^1^H NMR (500 MHz, CDCl_3_) δ 7.31 (dd, *J* = 7.3, 2.4 Hz, 2H), 7.18–7.13 (m, 3H), 7.12–7.07 (m, 1H), 6.91 (d, *J* = 7.4 Hz, 1H), 6.79 (t, *J* = 7.6 Hz, 1H), 6.61 (d, *J* = 10.5 Hz, 1H), 6.32 (dd, *J* = 10.4, 2.3 Hz, 1H), 5.30 (s, 1H), 4.92 (d, *J* = 2.3 Hz, 1H), 3.77 (d, *J* = 16.5 Hz, 1H), 2.79 (s, 6H), 2.59 (d, *J* = 16.5 Hz, 1H), 1.30 (s, 9H). ^13^C NMR (125 MHz, CDCl_3_) δ 195.9, 158.1, 152.6, 142.4, 137.9, 137.5, 127.7, 127.7, 127.6, 127.3, 124.5, 120.9, 120.4, 118.7, 97.6, 82.9, 50.0, 39.9, 38.3, 34.8, 29.9. HRMS (ESI): [M+Na]^+^ calcd. for C_26_H_29_NNaO_2_^+^: 410.2091, found: 410.2094.

8-(tert-butyl)-2-phenyl-4′-(pyrrolidin-1-yl)spiro[chromane-3,1′-cyclohexane]-2′,4′-dien-6′-one (**3m**). Flash column chromatography on a silica gel (ethyl acetate: petroleum ether, 1:3) afforded the product (17.7 mg, 43% yield) as a white solid.

^1^H NMR (500 MHz, CDCl_3_) δ 7.34 (dd, *J* = 7.5, 1.7 Hz, 2H), 7.20–7.15 (m, 3H), 7.11 (d, *J* = 7.5 Hz, 1H), 6.92 (d, *J* = 7.4 Hz, 1H), 6.80 (t, *J* = 7.6 Hz, 1H), 6.63 (d, *J* = 10.3 Hz, 1H), 6.23 (dd, *J* = 10.3, 1.9 Hz, 1H), 5.35 (s, 1H), 4.87 (d, *J* = 1.8 Hz, 1H), 3.78 (d, *J* = 16.5 Hz, 1H), 3.32 (m, 2H), 3.03 (m, 2H), 2.60 (d, *J* = 16.5 Hz, 1H), 1.82 (s, 4H), 1.31 (s, 9H). ^13^C NMR (125 MHz, CDCl_3_) δ 195.3, 155.7, 152.6, 142.7, 137.9, 137.7, 127.7, 127.7, 127.6, 127.3, 124.5, 121.0, 120.3, 119.9, 97.3, 82.8, 50.3, 48.0, 47.8, 38.6, 34.8, 29.9, 25.3, 24.6. HRMS (ESI): [M+Na]^+^ calcd. for C_28_H_31_NNaO_2_^+^: 436.2247, found: 436.2251.

8-(tert-butyl)-2-phenyl-4′-(piperidin-1-yl)spiro[chromane-3,1′-cyclohexane]-2′,4′-dien-6′-one (**3n**). Flash column chromatography on a silica gel (ethyl acetate: petroleum ether, 1:5) afforded the product (27.3 mg, 64% yield) as a white solid.

^1^H NMR (500 MHz, CDCl_3_) δ 7.42–7.36 (m, 2H), 7.24 (m, 3H), 7.19 (d, *J* = 7.7 Hz, 1H), 7.03 (d, *J* = 7.4 Hz, 1H), 6.89 (t, *J* = 7.6 Hz, 1H), 6.62 (d, *J* = 10.4 Hz, 1H), 6.40 (d, *J* = 10.4 Hz, 1H), 5.32 (s, 1H), 5.13 (s, 1H), 3.91 (d, *J* = 16.5 Hz, 1H), 3.24 (t, *J* = 5.2 Hz, 4H), 2.71 (d, *J* = 16.6 Hz, 1H), 1.63–1.54 (m, 2H), 1.44–1.32 (m, 13H). ^13^C NMR (125 MHz, CDCl_3_) δ 196.2, 157.4, 152.6, 141.6, 137.8, 137.1, 127.8, 127.6, 127.5, 127.1, 124.4, 120.8, 120.4, 119.1, 99.1, 83.4, 49.9, 48.2, 36.9, 34.7, 29.8, 25.2, 24.3. HRMS (ESI): calcd. for C_29_H_33_NO_2_ [M+Na]^+^: 450.2404, found: 450.2404. HRMS (ESI): [M+Na]^+^ calcd. for C_29_H_33_NNaO_2_^+^: 450.2404, found: 450.2408.

4′-(azepan-1-yl)-8-(tert-butyl)-2-phenylspiro[chromane-3,1′-cyclohexane]-2′,4′-dien-6′-one (**3o**). Flash column chromatography on a silica gel (ethyl acetate: petroleum ether, 1:3) afforded the product (26.9 mg, 61% yield) as a white solid.

^1^H NMR (500 MHz, CDCl_3_) δ 7.46–7.37 (m, 2H), 7.28–7.16 (m, 4H), 7.05 (dd, *J* = 7.5, 1.5 Hz, 1H), 6.91 (t, *J* = 7.6 Hz, 1H), 6.65 (d, *J* = 10.4 Hz, 1H), 6.41 (dd, *J* = 10.5, 2.3 Hz, 1H), 5.34 (s, 1H), 5.07 (d, *J* = 2.2 Hz, 1H), 3.95 (d, *J* = 16.5 Hz, 1H), 3.67–3.07 (m, 4H), 2.71 (d, *J* = 16.6 Hz, 1H), 1.58 (s, 4H), 1.40 (s, 9H), 1.28 (s, 4H). ^13^C NMR (125 MHz, CDCl_3_) δ 195.5, 157.2, 152.6, 142.2, 137.9, 137.2, 127.9, 127.7, 127.6, 127.1, 124.4, 120.9, 120.4, 118.8, 97.6, 83.5, 49.9, 36.9, 34.8, 29.9. HRMS (ESI): [M+Na]^+^ calcd. for C_30_H_35_NNaO_2_^+^: 464.2560, found: 464.2566.

8-(tert-butyl)-4′-(dibenzylamino)-2-phenylspiro[chromane-3,1′-cyclohexane]-2′,4′-dien-6′-one (**3p**). Flash column chromatography on a silica gel (ethyl acetate: petroleum ether, 1:3) afforded the product (38.3 mg, 71% yield) as a white solid.

^1^H NMR (500 MHz, CDCl_3_) δ 7.42 (m, 3H), 7.37–7.27 (m, 8H), 7.21 (d, *J* = 7.4 Hz, 1H), 7.06 (d, *J* = 7.3 Hz, 1H), 6.91 (m, 5H), 6.67 (d, *J* = 10.4 Hz, 1H), 6.43 (dd, *J* = 10.4, 2.1 Hz, 1H), 5.32 (s, 1H), 5.24 (d, *J* = 2.1 Hz, 1H), 4.41 (s, 4H), 3.96 (d, *J* = 16.6 Hz, 1H), 2.77 (d, *J* = 16.7 Hz, 1H), 1.40 (s, 9H). ^13^C NMR (100 MHz, CDCl_3_) δ 196.4, 158.3, 152.6, 142.9, 137.9, 137.1, 128.9, 127.9, 127.7, 127.6, 127.3, 126.3, 124.5, 120.7, 120.6, 119.2, 98.7, 83.8, 50.4, 36.4, 34.8, 29.9. HRMS (ESI): [M+Na]^+^ calcd. for C_38_H_37_NNaO_2_^+^: 562.2717, found: 562.2724.

8-(tert-butyl)-4′-(diallylamino)-2-phenylspiro[chromane-3,1′-cyclohexane]-2′,4′-dien-6′-one (**3q**). Flash column chromatography on a silica gel (ethyl acetate: petroleum ether, 1:3) afforded the product (36.4 mg, 83% yield) as a yellow oil.

^1^H NMR (500 MHz, CDCl_3_) δ 7.36 (m, 2H), 7.23 (m, 3H), 7.19 (d, *J* = 7.7 Hz, 1H), 7.03 (d, *J* = 7.5 Hz, 1H), 6.89 (t, *J* = 7.6 Hz, 1H), 6.62 (d, *J* = 10.4 Hz, 1H), 6.28 (dd, *J* = 10.4, 2.3 Hz, 1H), 5.73–5.51 (m, 2H), 5.29 (s, 1H), 5.11 (d, *J* = 10.4 Hz, 2H), 5.07 (d, *J* = 2.2 Hz, 1H), 4.81 (d, *J* = 17.2 Hz, 2H), 3.91 (d, *J* = 16.6 Hz, 1H), 3.72 (s, 4H), 2.70 (d, *J* = 16.7 Hz, 1H), 1.38 (s, 9H). ^13^C NMR (125 MHz, CDCl_3_) δ 196.1, 157.6, 152.5, 142.1, 137.8, 136.9, 127.8, 127.5, 127.4, 127.0, 124.4, 120.7, 120.4, 119.2, 117.1, 98.1, 83.5, 52.2, 50.0, 36.6, 34.7, 29.8. HRMS (ESI): [M+Na]^+^ calcd. for C_30_H_33_NO_2_^+^: 462.2404, found: 462.2409.

2-([1,1′-biphenyl]-4-yl)-8-(tert-butyl)-4′-(diallylamino)spiro[chromane-3,1′-cyclohexane]-2′,4′-dien-6′-one (**3r**). Flash column chromatography on a silica gel (ethyl acetate: petroleum ether, 1:3) afforded the product (43.3 mg, 84% yield) as a yellow oil.

^1^H NMR (500 MHz, CDCl_3_) δ 7.64–7.53 (m, 2H), 7.47 (d, *J* = 8.2 Hz, 2H), 7.50–7.39 (m, 4H), 7.37–7.31 (m, 1H), 7.20 (dd, *J* = 7.8, 1.7 Hz, 1H), 7.04 (dd, *J* = 7.6, 1.5 Hz, 1H), 6.89 (t, *J* = 7.6 Hz, 1H), 6.64 (d, *J* = 10.4 Hz, 1H), 6.30 (dd, *J* = 10.4, 2.3 Hz, 1H), 5.59 (s, 2H), 5.33 (s, 1H), 5.12–5.01 (m, 3H), 4.81– 4.77 (m, 2H), 3.93 (d, *J* = 16.4 Hz, 1H), 3.71 (s, 4H), 2.71 (d, *J* = 16.6 Hz, 1H), 1.39 (s, 9H). ^13^C NMR (125 MHz, CDCl_3_) δ 196.1, 157.7, 152.6, 142.2, 140.9, 140.3, 137.9, 136.2, 128.8, 127.9, 127.9, 127.2, 126.9, 125.8, 124.5, 120.8, 120.5, 119.4, 98.3, 83.4, 77.3, 50.2, 36.7, 34.8, 29.9. HRMS (ESI): [M+Na]^+^ calcd. for C_36_H_37_NNaO_2_^+^: 538.2717, found: 538.2715.

2-([1,1′-biphenyl]-4-yl)-8-(tert-butyl)-4′-(diethylamino)spiro[chromane-3,1′-cyclohexane]-2′,4′-dien-6′-one (**3s**). Flash column chromatography on a silica gel (ethyl acetate: petroleum ether, 1:3) afforded the product (40.7 mg, 83% yield) as a white oil.

^1^H NMR (500 MHz, CDCl_3_) δ 7.63–7.55 (m, 2H), 7.52–7.41 (m, 6H), 7.39–7.32 (m, 1H), 7.23 (dd, *J* = 7.7, 1.7 Hz, 1H), 7.07 (dd, *J* = 7.5, 1.6 Hz, 1H), 6.93 (t, *J* = 7.6 Hz, 1H), 6.69 (d, *J* = 10.4 Hz, 1H), 6.39 (dd, *J* = 10.5, 2.3 Hz, 1H), 5.39 (s, 1H), 5.07 (d, *J* = 2.3 Hz, 1H), 3.98 (dd, *J* = 16.5, 1.0 Hz, 1H), 3.19 (q, *J* = 7.0 Hz, 4H), 2.74 (d, *J* = 16.6 Hz, 1H), 1.43 (s, 9H), 1.01 (t, *J* = 7.0 Hz, 6H). ^13^C NMR (125 MHz, CDCl_3_) δ 195.3, 156.3, 152.6, 142.2, 141.0, 140.3, 137.9, 136.3, 128.8, 128.0, 127.9, 127.2, 126.9, 125.8, 124.5, 120.9, 120.5, 119.1, 97.5, 83.3, 49.9, 45.0, 36.8, 34.8, 29.9. HRMS (ESI): [M+Na]^+^ calcd. for C_34_H_37_NNaO_2_^+^: 514.2717, found: 514.2719.

4′-(dibenzylamino)-8-isopropyl-2-phenylspiro[chromane-3,1′-cyclohexane]-2′,4′-dien-6′-one (**3t**). Flash column chromatography on a silica gel (ethyl acetate: petroleum ether, 1:5) afforded the product (25.7 mg, 49% yield) as a yellow oil.

^1^H NMR (500 MHz, CDCl_3_) δ 7.41–7.35 (m, 3H), 7.34–7.25 (m, 8H), 7.13–7.08 (m, 1H), 6.99 (dd, *J* = 8.0, 1.3 Hz, 1H), 6.90 (t, *J* = 7.6 Hz, 1H), 6.88–6.83 (m, 4H), 6.56 (d, *J* = 10.5 Hz, 1H), 6.36 (dd, *J* = 10.5, 2.3 Hz, 1H), 5.31 (s, 1H), 5.23 (d, *J* = 2.3 Hz, 1H), 4.36 (d, *J* = 13.0 Hz, 4H), 3.90 (dd, *J* = 16.5, 1.2 Hz, 1H), 3.38–3.30 (m, 1H), 2.71 (d, *J* = 16.6 Hz, 1H), 1.22 (dd, *J* = 14.5, 6.9 Hz, 6H). ^13^C NMR (125 MHz, CDCl_3_) δ 196.5, 158.3, 150.8, 142.7, 137.3, 136.4, 128.9, 127.7, 127.6, 127.5, 127.3, 127.2, 126.3, 124.0, 120.7, 119.8, 119.2, 98.7, 83.5, 77.3, 50.3, 36.3, 26.9, 23.1, 22.3. HRMS (ESI): [M+Na]^+^ calcd. for C_37_H_35_NNaO_2_^+^: 548.2560, found: 548.2568.

8-(sec-butyl)-4′-(dibenzylamino)-2-phenylspiro[chromane-3,1′-cyclohexane]-2′,4′-dien-6′-one (**3u**). Flash column chromatography on a silica gel (ethyl acetate: petroleum ether, 1:5) afforded the product (36.1 mg, 67% yield, dr 2:1) as a yellow oil.

^1^H NMR (500 MHz, CDCl_3_) δ 7.41–7.34 (m, 3H), 7.35–7.20 (m, 8H), 7.09–7.04 (m, 1H), 7.02–6.94 (m, 1H), 6.98–6.89 (m, 1H), 6.87–6.80 (m, 4H), 6.55 (m, 1H), 6.39–6.30 (m, 1H), 5.31 (d, *J* = 3.1 Hz, 1H), 5.23 (t, *J* = 2.8 Hz, 1H), 4.58–4.23 (m, 4H), 3.96–3.84 (m, 1H), 3.16–3.09 (m, 1H), 2.71 (d, *J* = 16.6 Hz, 1H), 1.66–1.48 (m, 2H), 1.20 (m, 3H), 0.86 (t, *J* = 7.4 Hz, 2H), 0.79 (t, *J* = 7.4 Hz, 1H). ^13^C NMR (125 MHz, CDCl_3_) δ 196.6, 196.6, 158.4, 158.4, 151.2, 151.2, 142.9, 142.8, 137.3, 135.2, 135.2, 128.9, 128.6, 127.7, 127.6, 127.6, 127.5, 127.5, 127.3, 127.3, 127.1, 126.3, 124.8, 120.7, 119.8, 119.7, 119.2, 119.1, 98.7, 98.7, 83.6, 83.4, 77.3, 50.3, 36.3, 33.9, 33.3, 30.7, 29.5, 20.9, 20.3, 12.4, 12.2. HRMS (ESI): [M+Na]^+^ calcd. for C_38_H_37_NNaO_2_^+^: 562.2717, found: 562.2723.

8-(tert-butyl)-4′-(diethylamino)-2-ethylspiro[chromane-3,1′-cyclohexane]-2′,4′-dien-6′-one (**4k**). Flash column chromatography on a silica gel (ethyl acetate: petroleum ether, 1:3) afforded the product (30.1 mg, 82% yield, dr 1:1) as a white solid.

^1^H NMR (400 MHz, CDCl_3_) δ 7.13 (m, 2H), 6.93 (m, 2H), 6.79 (m, 2H), 6.02 (m, 1H), 5.64 (m, 1H), 5.30 (m, 1H), 5.21 (m, 2H), 5.04 (m, 1H), 4.37 (m, 1H), 3.44–3.21 (m, 7H), 3.19–3.02 (m, 2H), 2.75 (m, 2H), 2.42 (m, 4H), 2.01–1.93 (m, 2H), 1.78–1.59 (m, 4H), 1.41 (m, 16H), 1.18 (m, 13H). ^13^C NMR (100 MHz, CDCl_3_) δ 198.7, 163.1, 162.9, 137.1, 134.4, 128.1, 127.9, 124.3, 121.5, 121.3, 119.9, 119.3, 116.4, 98.5, 98.3, 82.3, 79.4, 44.2, 42.3, 42.0, 34.8, 34.7, 34.5, 34.0, 29.7, 29.6, 23.5, 22.8, 22.6, 22.6, 12.2. HRMS (ESI): [M+Na]^+^ calcd. for C_24_H_33_NNaO_2_^+^: 390.2404, found: 390.2411.

6,6′-((3-(tert-butyl)-2-((3,5-dimethoxybenzyl)oxy)phenyl)methylene)bis(3-(diethylamino)phenol) (**5a**). Flash column chromatography on a silica gel (ethyl acetate: petroleum ether, 1:15) afforded the product (51.8 mg, 81% yield) as a white solid.

^1^H NMR (500 MHz, CDCl_3_) δ 7.29 (m, 1H), 7.14 (dd, *J* = 7.7, 1.8 Hz, 1H), 7.09 (t, *J* = 7.7 Hz, 1H), 6.94 (d, *J* = 8.3 Hz, 2H), 6.56 (d, *J* = 2.4 Hz, 2H), 6.44 (t, *J* = 2.3 Hz, 1H), 6.22 (d, *J* = 7.9 Hz, 4H), 5.76 (s, 1H), 5.19 (s, 2H), 4.75 (s, 2H), 3.81 (s, 6H), 3.31 (q, *J* = 7.1 Hz, 8H), 1.50 (s, 9H), 1.15 (t, *J* = 7.0 Hz, 12H). ^13^C NMR (100 MHz, CDCl_3_) δ 160.9, 155.5, 148.6, 139.3, 135.4, 130.7, 129.4, 125.7, 124.4, 114.3, 105.2, 104.6, 100.3, 100.0, 76.6, 55.3, 44.3, 37.4, 35.4, 31.3, 12.7. HRMS (ESI): [M+Na]^+^ calcd. for C_40_H_52_N_2_NaO_5_^+^: 663.3768, found: 663.3778.

### 3.8. In Vitro Antifungal Activities

Each target compound was dissolved in acetone to prepare the stock solution (2.5 g/L). The stock solution was added into the PDA medium, and the concentration of target compound in the medium was 200.0 mg/L. Pure acetone without the target compound was utilized as the blank control, and difenoconazole and thifluzamide were co-assayed as the reference compound. Fresh dishes with a diameter of 6 mm were taken from the edge of the PDA-cultured fungi colonies and inoculated on the above three PDA media. Each treatment was tested for three replicates, and the antifungal effect was averaged. The relative inhibitory rate I (%) of all the tested compounds was calculated through the equation: I (%) = [(C − T)/(C − 5)] × 100. In this equation, I is the inhibitory rate and C and T are the colony diameter of the blank control (mm) and treatment (mm), respectively. Mycelia growth of four crop pathogenic fungi after treatment with the target compounds on PDA medium is illustrated in the Appendix A.

## 4. Conclusions

In conclusion, we have developed the Sc(OTf)_3_-catalyzed dearomative [5+1] annulation between readily available 3-aminophenols and *O*-alkyl *ortho*-oxybenzaldehydes for synthesis of spiro[chromane-3,1′-cyclohexane]-2′,4′-dien-6′-ones. The “two-birds-with-one-stone” strategy was disclosed by the dearomatization of phenols and direct α-C(sp^3^)–H bond functionalization of oxygen for providing a variety of spiro[chromane-3,1′-cyclohexane]-2′,4′-dien-6′-ones through cascade condensation/[1,5]-hydride transfer/dearomative-cyclization process. The antifungal activity assay and derivatizations of products were conducted to further enrich the utility of the structure. This work offers aromatization of the in situ-formed *o*-QM as hydride acceptor for initiating the hydride transfer-involved dearomatization, which would further enhance the utility of hydride transfer strategy in dearomatization chemistry.

## Data Availability

The data presented in this study are available on request from the corresponding authors.

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
