# Peer review of "Dearomatization of 3-Aminophenols for Synthesis of Spiro[chromane-3,1′-cyclohexane]-2′,4′-dien-6′-ones via Hydride Transfer Strategy-Enabled [5+1] Annulations"

_molecules, 2024, doi:10.3390/molecules29051012_

Round 1

Reviewer 1 Report

Comments and Suggestions for Authors

The manuscript entitled “Dearomatization of 3-Aminophenols for Synthesis of Cyclohexadienone Fused Spirochromanes via Hydride Transfer Strategy-enabled [5+1] Annulations” by Li and coworkers deals with scandium catalyzed dearomative annulation between 3-aminophenols and O-substituted benzaldehydes. The authors proposed a new strategy for synthesis of cyclohexadienone fused spirochromanes from readily available substrates. The authors present a reliable support for their mechanistic proposal in hydride transfer by investigation of steric repulsion between alkoxy and alkyl groups. Moreover, discovered antifungal activity of synthesized products highlights the developed protocol. The manuscript is well-written, and the supporting information contains all the necessary experimental procedures, characterization data of products, and spectra of synthesized compounds. Thus, I support publication after minor revision. The following minor details for improvement or consideration can be pointed out:

1.  Term “antimicrobial” should be changed to “antifungal” (Lines 16, 63, 134, 499).

2.  Term “large-scale synthesis” for such relatively simple substances when applied to 1 mmol scale seems to be not entirely correct. (lines 130, 144, 196).

3. Line 182 and p. S2(SI): 4.87 is a signal of water in methanol-d4, not methanol-d4 itself.

4. The reaction time should be indicated in Table 1, Schemes 2 and 3, and Chapter 3 “Materials and Methods”. “Upon completion of the reaction as indicated by TLC analysis” is too uncertain.

5. Lines 197, 198 and p. S2(SI): “O-alkyl ortho-oxybenzaldehydes 1a or 1k (1.0 mmol), phenol 2a (1.0 mmol)”: Optimal ratio of reagents 1 and 2 is 1:1.5 (Table 1, entry 21). The same ratio 1:1.5 is described in Scheme p. S2 (SI) for large-scale synthesis.

6. Lines 213, 214, 471 and p. S3(SI): “O-alkyl ortho-oxybenzaldehyde 1v (0.1 mmol), phenol 2a (0.15 mmol)”: With this ratio of reagents it is impossible to obtain yield 5a of 51.8 mg (81%).

7. There are some errors in the description of NMR spectra (Chapter 3 and SI):

Line 246: 7.30 (dd, J = 6.4, 2.8 Hz, 2H) – These coupling constants cannot exist in this compound; should be changed to m.

Line 247: 7.02 (dd, J = 14.6, 7.4 Hz, 2H) These coupling constants cannot exist in this compound; should be changed to m.

Line 277: 7.16 (d, J = 8.0 Hz, 3H) This coupling constant cannot exist for 3H; should be changed to m.

Line 287: 7.29 (d, J = 1.9 Hz, 1H), 7.27 (s, 1H) These signals probably contain a chloroform signal. There should be a doublet of two protons of the 4-methylphenyl substituent, J ca. 8 Hz.

Lines 287-288: 7.04 (d, J = 7.8 Hz, 3H) This coupling constant cannot exist for 3H; should be changed to m.

Lines 368-369: 3.32 (d, J = 34.9 Hz, 2H), 3.03 (d, J = 40.6 Hz, 2H) These coupling constants cannot exist!! should be changed to m.

Line 376: 7.24 (d, J = 4.9 Hz, 3H) This coupling constant cannot exist in this compound; should be changed to m.

Line 396: 7.42 (d, J = 6.9 Hz, 3H) This coupling constant cannot exist for 3H; should be changed to m.

Line 397: 6.91 (dd, J = 12.8, 7.3 Hz, 5H) This dd cannot exist in this compound; should be changed to m.

Line 406: 7.36 (dd, J = 6.8, 2.8 Hz, 2H), 7.23 (dd, J = 5.0, 1.7 Hz, 3H) These coupling constants cannot exist in this compound; should be changed to m.

Line 449, 451-452: 6.55 (t, J = 10.1 Hz, 1H)- not t, but doublet for each isomer. Also 1.20 (t, J = 6.8 Hz, 3H) is not t, but two doublets of the Me group. These signals should be changed to m.

Lines 460-465: The compound is a mixture of isomers and all 1H NMR signals should be changed to m. In addition, the 1H NMR spectrum contains too many protons.

Line 472: There are 53 protons in the spectrum, but there should be 52. Signal 7.29 (dd, J = 7.8, 1.8 Hz, 2H) probably contains the signal of chloroform.

Author Response

Reviewer: 1

Recommendation: Publish after minor revisions noted.

Comments:

The manuscript entitled “Dearomatization of 3-Aminophenols for Synthesis of Cyclohexadienone Fused Spirochromanes via Hydride Transfer Strategy-enabled [5+1] Annulations” by Li and coworkers deals with scandium catalyzed dearomative annulation between 3-aminophenols and O-substituted benzaldehydes. The authors proposed a new strategy for synthesis of cyclohexadienone fused spirochromanes from readily available substrates. The authors present a reliable support for their mechanistic proposal in hydride transfer by investigation of steric repulsion between alkoxy and alkyl groups. Moreover, discovered antifungal activity of synthesized products highlights the developed protocol. The manuscript is well-written, and the supporting information contains all the necessary experimental procedures, characterization data of products, and spectra of synthesized compounds. Thus, I support publication after minor revision. The following minor details for improvement or consideration can be pointed out:

  1. Term “antimicrobial” should be changed to “antifungal” (Lines 16, 63, 134, 499).

Reply: Thank you for your suggestions. The term “antimicrobial” has been changed by “antifungal” in the revised manuscript.

  1. Term “large-scale synthesis” for such relatively simple substances when applied to 1 mmol scale seems to be not entirely correct. (lines 130, 144, 196).

Reply: Thank you for your suggestions. In order to demonstrate the robustness of the reaction, the gram-scale synthesis of 3a and 3k was conducted. And 4 mmol scale synthesis gave 3a and 3k in 67% and 74% yields, respectively. The term “large-scale synthesis” has been changed by “gram-scale synthesis” in the revised manuscript.

  1. Line 182 and p. S2(SI): 4.87 is a signal of water in methanol-d4, not methanol-d4 itself.

Reply: Thank you so much for your reminding. The relevant data have been added in the revised manuscript [methanol-d4 (δ 3.31, quintuplet)].

  1. The reaction time should be indicated in Table 1, Schemes 2 and 3, and Chapter 3 “Materials and Methods”. “Upon completion of the reaction as indicated by TLC analysis” is too uncertain.

Reply: Thank you so much for your reminding. The reaction time has been indicated in Table 1, Schemes 2 and 3, and Chapter 3 “Materials and Methods” and added in the revised manuscript.

  1. Lines 197, 198 and p. S2(SI): “O-alkyl ortho-oxybenzaldehydes 1a or 1k (1.0 mmol), phenol 2a (1.0 mmol)”: Optimal ratio of reagents 1 and 2 is 1:1.5 (Table 1, entry 21). The same ratio 1:1.5 is described in Scheme p. S2 (SI) for large-scale synthesis.

Reply: Thank you so much for your reminding. It is our careless. The ratio of reagents 1 and 2 is 1:1.5. And We have carefully revised these errors in the revised manuscript.

  1. Lines 213, 214, 471 and p. S3(SI): “O-alkyl ortho-oxybenzaldehyde 1v (0.1 mmol), phenol 2a (0.15 mmol)”: With this ratio of reagents it is impossible to obtain yield 5a of 51.8 mg (81%).

Reply: Thank you so much for your reminding. It is our careless. The ratio of O-alkyl ortho-oxybenzaldehyde 1v (0.1 mmol) and phenol 2a (0.22 mmol) is 1:2.2. We have carefully revised this error in the revised manuscript.

  1. There are some errors in the description of NMR spectra (Chapter 3 and SI):

Line 246: 7.30 (dd, J = 6.4, 2.8 Hz, 2H) – These coupling constants cannot exist in this compound; should be changed to m.

Line 247: 7.02 (dd, J = 14.6, 7.4 Hz, 2H) These coupling constants cannot exist in this compound; should be changed to m.

Line 277: 7.16 (d, J = 8.0 Hz, 3H) This coupling constant cannot exist for 3H; should be changed to m.

Line 287: 7.29 (d, J = 1.9 Hz, 1H), 7.27 (s, 1H) These signals probably contain a chloroform signal. There should be a doublet of two protons of the 4-methylphenyl substituent, J ca. 8 Hz.

Lines 287-288: 7.04 (d, J = 7.8 Hz, 3H) This coupling constant cannot exist for 3H; should be changed to m.

Lines 368-369: 3.32 (d, J = 34.9 Hz, 2H), 3.03 (d, J = 40.6 Hz, 2H) These coupling constants cannot exist!! should be changed to m.

Line 376: 7.24 (d, J = 4.9 Hz, 3H) This coupling constant cannot exist in this compound; should be changed to m.

Line 396: 7.42 (d, J = 6.9 Hz, 3H) This coupling constant cannot exist for 3H; should be changed to m.

Line 397: 6.91 (dd, J = 12.8, 7.3 Hz, 5H) This dd cannot exist in this compound; should be changed to m.

Line 406: 7.36 (dd, J = 6.8, 2.8 Hz, 2H), 7.23 (dd, J = 5.0, 1.7 Hz, 3H) These coupling constants cannot exist in this compound; should be changed to m.

Line 449, 451-452: 6.55 (t, J = 10.1 Hz, 1H)- not t, but doublet for each isomer. Also 1.20 (t, J = 6.8 Hz, 3H) is not t, but two doublets of the Me group. These signals should be changed to m.

Lines 460-465: The compound is a mixture of isomers and all 1H NMR signals should be changed to m. In addition, the 1H NMR spectrum contains too many protons.

Line 472: There are 53 protons in the spectrum, but there should be 52. Signal 7.29 (dd, J = 7.8, 1.8 Hz, 2H) probably contains the signal of chloroform.

Reply: Thank you so much for your reminding. It is our careless. We have carefully revised these errors in the manuscript. The revised manuscript was checked again by the authors to make sure there were no such issues.

Reviewer 2 Report

Comments and Suggestions for Authors

In this manuscript, Li and coworkers describe the development of a Sc(OTf)3-catalyzed dearomative [5+1] annulation reaction between 3-aminophenols and O-alkyl ortho-oxybenzaldehydes, resulting in the synthesis of differently substituted 4'-aminospiro[chromane-3,1'-cyclohexane]-2',4'-dien-6'-one. The authors illustrate the efficacy, versatility, and mildness of the methodology by successfully synthesizing widely substituted aminospiro[chromane-cyclohexane]dieneone in high yields, even in the presence of a few sensitive functional groups. Based on evidence from controlled experiments, the authors propose a plausible mechanism involving Sc-catalyzed condensation of aminophenols with O-alkyl ortho-oxybenzaldehydes, followed by functionalization of the oxygen-attached C(sp3)–H bond via [1,5]-hydride transfer and dearomative cyclization to produce the final product. The authors also conducted in vitro antifungal assays on the synthesized products to demonstrate their potential utility.

While several dearomative cyclizations of differently substituted phenols catalyzed by Sc(OTf)3 have been reported, this manuscript introduces a new synthetic strategy for constructing differently substituted 4'-aminospiro[chromane-3,1'-cyclohexane]-2',4'-dien-6'-one through dearomative [5+1] annulations. Given the pharmacological importance of this complex spirocyclic scaffold with chiral heterocyclic rings and fused aromatic rings, the methodology's single-step integration of dearomatization of phenols, functionalization of oxygen-attached C(sp3)–H bond, [1,5]-hydride transfer, and dearomative cyclization holds significance in organic synthesis. However, the manuscript has several issues that needs to addressed before it can be considered for publication in Molecules.

1.     The title and throughout the manuscript refer to the synthesized system as ‘Cyclo-hexadienone Fused Spirochromanes.’ The terms 'fused' and 'spiro' scientifically express two different bicyclic systems. Therefore, using 'fused' to describe a 'spiro' system is inaccurate and confusing. The authors need to correct this term in all sentences throughout the manuscript.

2.     Spiro[chromane-3,1'-cyclohexane]-2',4'-dien-6'-one and spiro[chromane-3,1'-cyclohexane]-2',5'-dien-4'-one have different chemical structures and are expected to exhibit different chemical reactivity and biological activity. As the manuscript deals only with the former, specifying the system in the title or abstract is important for improving content accuracy.

3.     Although the abstract and conclusion mention the antimicrobial activity assay of the synthesized compounds and derivatives, the text only discusses the in vitro antifungal activity assay. To present accurate information in the abstract and draw the correct conclusion, the authors should replace the term 'antimicrobial' with 'antifungal.'

4.     In the main text, the authors have not discussed the bioassay result and have only presented the procedure used for the antifungal activity assay. The authors must discuss the bioassay result to justify its inclusion in the abstract and conclusion sections.

5.     Some parts of the manuscript are well-written, especially the mechanistic exploration section, while others lack attention to detail, with crucial information missing. For example, the optimization table lacks a column or mention of reaction time, and the general procedure is missing the reaction time. In every relevant section, authors must provide reaction time and other pertinent data.

6.     Authors have expressed the amounts of all substrates and reagents only in millimoles. Wherever possible, they should also provide the amounts of these substrates or reagents in terms of their actual weight (mg or g) in the same bracket.

7.     Even in the large-scale synthesis, authors have worked on a 1 mmol scale. Authors should show at least one reaction at the gram level of substrate and reagents to demonstrate the robustness of the reaction.

Author Response

Reviewer: 2

Recommendation: Publish after minor revisions noted.

Comments:

In this manuscript, Li and coworkers describe the development of a Sc(OTf)3-catalyzed dearomative [5+1] annulation reaction between 3-aminophenols and O-alkyl ortho-oxybenzaldehydes, resulting in the synthesis of differently substituted 4'-aminospiro[chromane-3,1'-cyclohexane]-2',4'-dien-6'-one. The authors illustrate the efficacy, versatility, and mildness of the methodology by successfully synthesizing widely substituted aminospiro[chromane-cyclohexane]dieneone in high yields, even in the presence of a few sensitive functional groups. Based on evidence from controlled experiments, the authors propose a plausible mechanism involving Sc-catalyzed condensation of aminophenols with O-alkyl ortho-oxybenzaldehydes, followed by functionalization of the oxygen-attached C(sp3)–H bond via [1,5]-hydride transfer and dearomative cyclization to produce the final product. The authors also conducted in vitro antifungal assays on the synthesized products to demonstrate their potential utility.

While several dearomative cyclizations of differently substituted phenols catalyzed by Sc(OTf)3 have been reported, this manuscript introduces a new synthetic strategy for constructing differently substituted 4'-aminospiro[chromane-3,1'-cyclohexane]-2',4'-dien-6'-one through dearomative [5+1] annulations. Given the pharmacological importance of this complex spirocyclic scaffold with chiral heterocyclic rings and fused aromatic rings, the methodology's single-step integration of dearomatization of phenols, functionalization of oxygen-attached C(sp3)–H bond, [1,5]-hydride transfer, and dearomative cyclization holds significance in organic synthesis. However, the manuscript has several issues that needs to addressed before it can be considered for publication in Molecules.

  1. The title and throughout the manuscript refer to the synthesized system as ‘Cyclo-hexadienone Fused Spirochromanes.’ The terms 'fused' and 'spiro' scientifically express two different bicyclic systems. Therefore, using 'fused' to describe a 'spiro' system is inaccurate and confusing. The authors need to correct this term in all sentences throughout the manuscript.

Reply: Thank you for your suggestions. The term “Cyclo-hexadienone Fused Spirochromanes” has been changed by “spiro[chromane-3,1’-cyclohexane]-2’,4’-dien-6’-one” in the revised manuscript. We have corrected this term in all sentences throughout the manuscript.

  1. Spiro[chromane-3,1'-cyclohexane]-2',4'-dien-6'-one and spiro[chromane-3,1'-cyclohexane]-2',5'-dien-4'-one have different chemical structures and are expected to exhibit different chemical reactivity and biological activity. As the manuscript deals only with the former, specifying the system in the title or abstract is important for improving content accuracy.

Reply: Thank you for your suggestions. As you mentioned, the term “Cyclo-hexadienone Fused Spirochromanes” has been changed by “spiro[chromane-3,1’-cyclohexane]-2’,4’-dien-6’-one” in the revised manuscript to improve content accuracy.

  1. Although the abstract and conclusion mention the antimicrobial activity assay of the synthesized compounds and derivatives, the text only discusses the in vitro antifungal activity assay. To present accurate information in the abstract and draw the correct conclusion, the authors should replace the term 'antimicrobial' with 'antifungal.'

Reply: Thank you for your suggestions. The term “antimicrobial” has been changed by “antifungal” in the revised manuscript.

  1. In the main text, the authors have not discussed the bioassay result and have only presented the procedure used for the antifungal activity assay. The authors must discuss the bioassay result to justify its inclusion in the abstract and conclusion sections.

Reply: Thank you for your suggestions. We have discussed the bioassay result, and the corresponding sentences have been added the revised manuscript.

Compounds 3a, 3j, 3k, 3q, 3t and 3u were evaluated for their antifungal activities against four economically important phytopathogenic fungi Rhizoctonia solani, Alternaria solani, Alternaria mali, and Sclerotium rolfsii. The results showed that most of the tested compounds possessed in vitro antifungal activity at a concentration of 200 mg/L. Especially, compound 3a exhibited remarkable antifungal potency among all target compounds, with inhibition rate of 40.21, 60.35, 53.56 and 29.09% at a concentration of 200 mg/L, respectively, against Rhizoctonia solani, Alternaria solani, Alternaria mali, and Sclerotium rolfsii.

  1. Some parts of the manuscript are well-written, especially the mechanistic exploration section, while others lack attention to detail, with crucial information missing. For example, the optimization table lacks a column or mention of reaction time, and the general procedure is missing the reaction time. In every relevant section, authors must provide reaction time and other pertinent data.

Reply: Thank you so much for your reminding. We have added the reaction time column to the optimization table. The reaction time and other pertinent data have been added in every relevant section.

  1. Authors have expressed the amounts of all substrates and reagents only in millimoles. Wherever possible, they should also provide the amounts of these substrates or reagents in terms of their actual weight (mg or g) in the same bracket.

Reply: Thank you for your suggestions. We have added the amounts of substrates or reagents in terms of their actual weight in the same bracket.

  1. Even in the large-scale synthesis, authors have worked on a 1 mmol scale. Authors should show at least one reaction at the gram level of substrate and reagents to demonstrate the robustness of the reaction.

Reply: Thank you for your suggestions. In order to demonstrate the robustness of the reaction, the gram-scale synthesis of 3a and 3k was conducted. And 4 mmol scale synthesis gave 3a and 3k in 67% and 74% yields, respectively. The term “large-scale synthesis” has been changed by “gram-scale synthesis” in the revised manuscript.

Round 2

Reviewer 2 Report

Comments and Suggestions for Authors

The authors have implemented the suggested changes to revise the manuscript, and it can now be accepted for publication in Molecules.